# Mitochondrial Dysfunction and Metabolic Reprogramming in Chronic Inflammatory Diseases: Molecular Insights and Therapeutic Opportunities

**DOI:** 10.3390/cimb47121042

**Published:** 2025-12-14

**Authors:** Mi Eun Kim, Yeeun Lim, Jun Sik Lee

**Affiliations:** Department of Biological Science, Immunology Research Lab, Chosun University, Gwangju 61452, Republic of Korea; kimme0303@chosun.ac.kr (M.E.K.); dpdms1403@naver.com (Y.L.)

**Keywords:** mitochondrial dysfunction, metabolic reprogramming, immuno-metabolism, chronic inflammation, therapeutic targeting

## Abstract

Chronic inflammatory diseases are driven by persistent immune activation and metabolic imbalance that disrupt tissue homeostasis. Mitochondrial dysfunction disrupts cellular bioenergetics and immune regulation, driving persistent inflammatory signaling. Mitochondrial dysfunction, characterized by excessive production of ROS, release of mitochondrial DNA, and defective mitophagy, amplifies inflammatory signaling and contributes to disease progression. Meanwhile, metabolic reprogramming in immune and stromal cells establishes distinct bioenergetic profiles. These profiles maintain either pro-inflammatory or anti-inflammatory phenotypes through key signaling regulators such as HIF-1α, AMPK, mTOR, and SIRT3. Crosstalk between mitochondrial and metabolic pathways determines whether inflammation persists or resolves. Recent advances have identified critical molecular regulators, including the NRF2–KEAP1 antioxidant system, the cGAS–STING innate immune pathway, and the PINK1–Parkin mitophagy pathway, as potential therapeutic targets. Pharmacologic modulation of metabolic checkpoints and restoration of mitochondrial homeostasis represent key strategies for re-establishing cellular homeostasis. Developing approaches, including NAD^+^ supplementation, mitochondrial transplantation, and gene-based interventions, also show significant therapeutic potential. This review provides a mechanistic synthesis of how mitochondrial dysfunction and metabolic reprogramming cooperate to maintain chronic inflammation and highlights molecular pathways that represent promising targets for precision therapeutics in inflammatory diseases.

## 1. Introduction

Chronic inflammatory diseases such as rheumatoid arthritis, systemic lupus erythematosus, and chronic obstructive pulmonary disease manifest prolonged immune activation that drives progressive tissue damage. Although conventional therapies targeting cytokines and immune signaling pathways have provided clinical benefits, many patients continue to experience incomplete resolution and recurrent disease activity. To address current limitations, we review emerging molecular pathways underlying chronic inflammation and discuss targeted strategies to reconstitute cellular balance. In recent years, the understanding of inflammation has evolved to include mechanisms that are not limited to immune receptor activation and cytokine signaling. Increasing evidence indicates that cellular metabolism and cellular compartmentalization play essential roles in regulating inflammatory responses. Among intracellular compartments, mitochondria uniquely integrate metabolic and inflammatory signals. Mitochondrial dysfunction can impair these homeostatic processes, resulting in excessive production of reactive oxygen species (ROS), release of mitochondrial DNA (mtDNA) into the cytoplasm, and activation of pro-inflammatory signaling pathways such as NF-κB and NLRP3 inflammasome. These events amplify inflammatory cascades and contribute to promoted tissue damage in chronic inflammatory diseases [1]. In similar, metabolic reprogramming has been recognized as a hallmark of immune cell activation and chronic inflammation. Activated macrophages, dendritic cells, and T lymphocytes undergo profound metabolic reprogramming to support cytokine synthesis, proliferation, and effector functions. Pro-inflammatory macrophages are driven predominantly on glycolysis, whereas regulatory or anti-inflammatory phenotypes depend on oxidative phosphorylation and fatty acid oxidation. Similar metabolic alterations are observed in stromal and epithelial cells within inflammatory tissue regions, suggesting that metabolic dysregulation is not limited to immune cells but extends to the tissue microenvironment [2]. These findings have led to increasing recognition that mitochondrial dysfunction and metabolic dysregulation are not simply consequences of inflammation but key drivers of disease progression. Understanding the molecular links between mitochondrial impairment, cellular metabolism, and inflammatory signaling is essential for identifying novel therapeutic targets. The objective of this review is to provide an integrated overview of how mitochondrial dysfunction and metabolic reprogramming contribute to chronic inflammatory diseases and to discuss emerging therapeutic strategies that aim to reconstitute mitochondrial homeostasis and metabolic balance. Therefore, we focus on how mitochondrial dysfunction and metabolic reprogramming drive chronic inflammation and evaluate their potential as therapeutic targets in precision medicine. The key interactions between mitochondrial dysfunction, metabolic alteration, immune cell responses, and downstream tissue injury are summarized in Figure 1.

## 2. Mitochondrial Dysfunction in Chronic Inflammation

Mitochondria function as the primary site of oxidative phosphorylation and ATP generation, but their role extends far beyond bioenergetics [3,4]. In chronic inflammatory diseases, mitochondria play a central role in driving inflammation when homeostatic mechanisms are compromised. This section examines three major mechanistic pathways by which mitochondrial dysfunction contributes to chronic inflammation: (i) excessive production of ROS and redox imbalance; (ii) release of mtDNA and activation of innate immune signaling; and (iii) dysregulation of mitochondrial processes and mitophagy leading to accumulation of damaged mitochondria and consequent inflammatory signaling [5,6,7,8].

### 2.1. ROS and Redox Imbalance

Mitochondria produce low levels of ROS, mainly superoxide and hydrogen peroxide, as byproducts of electron transport, which serve physiological redox signaling functions. Under stress conditions such as nutrient overload, hypoxia, or defective electron transport chain (ETC) components, mitochondrial ROS production exceeds the capacity of antioxidant defenses, resulting in oxidative stress and disruption of redox homeostasis [9,10,11]. Activation of the NRF2–KEAP1 pathway and mitochondrial antioxidant enzymes such as superoxide dismutase 2 (SOD2), peroxiredoxin-3 (PRDX3), and glutathione peroxidase 4 (GPX4) plays an essential role in limiting oxidative stress and preserving redox balance [12,13]. Chronic mitochondrial ROS accumulation induces oxidative modifications that activate NF-κB–dependent inflammatory signaling, thereby inducing pro-inflammatory cytokines including tumor necrosis factor alpha (TNF-α) and interleukin (IL)-1β [14,15]. Oxidative modification of mitochondrial membranes, particularly cardiolipin oxidation, destabilizes the inner mitochondrial membrane, leading to depolarization and decreased ATP synthesis. The resulting metabolic impairment accelerates ROS generation, forming a self-perpetuating cycle that maintains the inflammatory response [11,16,17,18]. Mitochondrial ROS also act as activators of the NLR family pyrin domain containing 3 (NLRP3) inflammasome, a multiprotein complex that triggers caspase-1 activation and cytokine maturation. Experimental inhibition of mitochondrial complexes I or III has been shown to increase mitochondrial ROS and induce NLRP3 activation in macrophages [19]. Consequently, redox imbalance serves as a critical molecular bridge between metabolic stress and chronic inflammation, suggesting that reconstitution of mitochondrial antioxidant systems could provide therapeutic potential. In addition to oxidative stress, a contrasting redox alteration described as reductive stress can also impair mitochondrial signaling. Increased NADH with reduced NAD^+^ limits electron flow through the respiratory chain and suppresses physiological ROS signals required for immune regulation. Recent studies report that reductive stress aggravates metabolic inflammation and promotes mitochondrial dysfunction in chronic disease [20,21].

### 2.2. Mitochondrial DNA Release and Innate Immune Activation

In addition to ROS production, mitochondria contain their own genome in the form of circular mtDNA, which encodes essential components of the oxidative phosphorylation system. Under normal conditions, mtDNA is tightly packaged within the mitochondrial matrix by mitochondrial transcription factor A (TFAM). Mitochondrial damage allows mtDNA leakage into the cytosol, triggering innate immune receptors such as cyclic GMP-AMP synthase (cGAS) and toll-like receptor (TLR)-9 [22,23,24]. This process is tightly regulated by the signaling pathway comprising TFAM, cGAS, and stimulator of interferon genes (STING), which senses cytosolic mtDNA and activates type I interferon and NF-κB responses. Dysregulation of this pathway contributes to prolonged inflammatory signaling in chronic disease conditions [25,26]. Oxidized mtDNA has been shown to bind directly to NLRP3, leading to caspase-1 activation and release of IL-1β and IL-18 [27,28]. The mechanisms underlying mtDNA release include mitochondrial outer membrane permeabilization by proapoptotic proteins Bax and Bak, the formation of pores through voltage-dependent anion channels (VDAC), and reduced degradation of damaged mitochondria due to impaired mitophagy [29]. Once present in the cytoplasm, mtDNA engages receptors such as TLR-9 and cGAS, which subsequently activate the STING pathway. This triggers downstream type I interferon production and NF-κB signaling, both of which support inflammatory responses [30]. Increased circulating mtDNA levels have been reported in patients with autoimmune diseases, sepsis, and metabolic inflammation, correlating with disease severity and mortality. In endothelial cells exposed to hyperglycemia or oxidative stress, extracellular mtDNA enhances inflammatory activation and vascular dysfunction [23]. Therefore, the release of mtDNA from damaged mitochondria provides a mechanistic association between cellular injury and innate immune activation, maintaining a chronic inflammatory response.

### 2.3. Mitochondrial Dynamics and Mitophagy

Mitochondria maintain their integrity through continuous cycles of fission and fusion, processes that allow quality control and adaptation to metabolic demands. Mitochondrial fusion, mediated by mitofusin 1 and 2 (MFN1 and MFN2) and optic atrophy 1 (OPA1), supports maintenance of cristae integrity and preserves respiratory efficiency. In contrast, dysregulated mitochondrial fission driven by dynamin-related protein 1 (DRP1) promotes fragmentation, increases ROS, and contributes to metabolic and inflammatory dysfunction. The balance of these contrasting processes ensures mitochondrial and cellular homeostasis. Alteration of this balance has been linked to various inflammatory disorders. Increased mitochondrial fission results in fragmented mitochondria that exhibit reduced ATP production and increased ROS levels. Phosphorylation of DRP1 at Ser616 in the human isoform enhances its recruitment to mitochondria and promotes fission, a process associated with increased IL-1β generation during inflammatory signaling. In contrast, phosphorylation at Ser637 restrains DRP1 activity through PKA and Ca^2+^-dependent pathways. Reduced Ser637 phosphorylation, together with increased Ser616 phosphorylation, promotes DRP1 activation under metabolic and inflammatory conditions [31,32].

Mitophagy, a selective form of autophagy, eliminates damaged mitochondria through a process orchestrated by PTEN-induced kinase 1 (PINK1) and the E3 ubiquitin ligase Parkin. Dysfunctions in the PINK1/Parkin pathway or BNIP3/NIX-mediated mitophagy impair mitochondrial clearance, promoting accumulation of dysfunctional mitochondria and prolonged NLRP3 inflammasome activation [33,34]. When mitophagy is impaired, dysfunctional mitochondria accumulate and continuously release ROS and DAMPs, leading to prolonged NLRP3 inflammasome activation and cytokine release [35,36]. Experimental models have shown that impaired mitophagy in macrophages or adipocytes contributes to chronic inflammation, which is a hallmark of obesity, atherosclerosis, and type 2 diabetes [37,38,39]. Altered mitochondrial processes drive cells toward glycolysis, linking morphological stress to metabolic inflammation. Conversely, enhanced fusion supports oxidative metabolism and anti-inflammatory cell polarization. This relationship between mitochondrial morphology and metabolic characteristics highlights the interaction between mitochondrial dysfunction and immune-metabolic regulation in chronic inflammation. Taken together, abnormalities in mitochondrial function and insufficient clearance of damaged mitochondria promote signaling pathways that associate mitochondrial injury with chronic inflammation. Re-establishing balanced mitochondrial processes and promoting efficient mitophagy may therefore represent promising therapeutic approaches to reduce chronic inflammatory diseases. The mitochondrial mechanisms contributing to chronic inflammation are summarized in Table 1.

Mitochondrial dysfunction alters energy output, redox balance, and metabolite availability, which together modify immune activity in chronic disease. These changes identify biochemical and cellular targets that can be used for therapeutic development. Approaches that regulate glycolysis, TCA cycle, lipid use, NAD^+^ levels, fusion–fission control, or ROS regulation are derived from these mechanistic findings. Mitochondria can move between immune cells and stromal cells through tunneling nanotubes or extracellular vesicles. This exchange can increase ATP levels when healthy mitochondria are transferred, or spread metabolic stress when damaged mitochondria are moved. Such transfer has been reported in inflammatory and injury models, where it influences tissue injury and recovery. These findings suggest that mitochondrial transfer contributes to disease processes and may have therapeutic potential [40,41,42].

## 3. Metabolic Reprogramming in Immune and Non-Immune Cells

Metabolic reprogramming describes the adaptive change in cellular energy utilization and biosynthetic pathways that occurs in response to environmental and immunological stimuli. In chronic inflammatory diseases, both immune and non-immune cells exhibit metabolic alterations that support the production of inflammatory mediators, cellular proliferation, and survival. These changes are not secondary effects but active determinants of immune cell phenotype and function. Metabolic reprogramming affects glycolysis, oxidative phosphorylation (OXPHOS), fatty acid oxidation (FAO), amino acid metabolism, and the tricarboxylic acid (TCA) cycle. Through these metabolic pathways, cells integrate environmental signals into functional responses that promote chronic inflammation. Consistent with these metabolic influences, several TCA-derived metabolites act on chromatin regulators to modify inflammatory gene activity. TCA-derived metabolites exert regulatory influence on chromatin enzymes and protein modification systems. Acetyl-CoA availability affects histone acetylation, enabling transcriptional programs linked to inflammatory activity. α-Ketoglutarate serves as a cofactor for dioxygenases that adjust histone and DNA methylation, thereby modifying immune cell function in chronic disease. Altered levels of these metabolites have been reported in arthritis, metabolic disorders, and neuroinflammatory diseases, indicating that metabolic–epigenetic interactions contribute to prolonged inflammatory responses [43,44].

Ferroptosis is an iron-dependent form of regulated cell injury driven by lipid peroxidation and impaired glutathione metabolism. Mitochondrial activity influences ferroptosis through changes in TCA activity, NADH–NAD^+^ balance, and production of lipid-derived ROS. Recent studies show that ferroptosis promotes release of pro-inflammatory mediators and contributes to chronic inflammatory disorders, including arthritis and metabolic disease. Inhibition of GPX4 activity or accumulation of polyunsaturated phospholipids increases susceptibility to ferroptosis and strengthens interactions between mitochondrial metabolism and immune activation [45,46].

### 3.1. Immuno-Metabolic Reprogramming in Macrophages and T Lymphocytes

Macrophages and T cells are key effector cells in chronic inflammation. Their activation phenotypes are intimately linked to defined metabolic pathways. Characteristically activated macrophages, known as M1 macrophages, depend primarily on glycolysis, even in the presence of sufficient oxygen, a phenomenon similar to the Warburg effect observed in tumor cells. This metabolic configuration allows rapid generation of ATP and biosynthetic precursors needed for cytokine production and phagocytosis. In contrast, alternatively activated M2 macrophages depend on oxidative phosphorylation and fatty acid oxidation, which support tissue repair and anti-inflammatory functions [2]. The transition from an inactive to an activated macrophage involves a reprogramming in key metabolic checkpoints. Upon stimulation by lipopolysaccharide (LPS) or interferon (IFN)-γ, the enzyme pyruvate dehydrogenase (PDH) becomes inhibited, diverting pyruvate away from the mitochondria and into lactate production. This causes accumulation of intermediates such as succinate, which stabilizes hypoxia-inducible factor (HIF)-1α and promotes transcription of IL-1β, an essential cytokine in inflammatory responses. This cascade is administrated by the HIF-1α/succinate feedback loop, a key checkpoint connecting hypoxia-induced signaling with pro-inflammatory metabolic remodeling [47,48]. Succinate also drives mitochondrial ROS generation, further enhancing inflammation. Conversely, M2 polarization induced by IL-4 promotes activation of peroxisome proliferator-activated receptor gamma coactivator-1β (PGC-1β), which enhances fatty acid oxidation and mitochondrial formation, promoting an anti-inflammatory phenotype [49,50]. The AMPK, PGC-1β, and PPARγ regulatory mechanism underlies this metabolic transition and associates fatty acid oxidation to anti-inflammatory macrophage differentiation [51,52].

T lymphocytes undergo similar metabolic reprogramming during activation and differentiation. Naïve T cells depend primarily on OXPHOS and FAO to maintain homeostasis. Upon antigen stimulation, effector T cells switch to glycolysis to rapidly generate ATP and biosynthetic substrates for proliferation and cytokine secretion. In contrast, regulatory T cells (Tregs) and memory T cells maintain a support on OXPHOS and FAO, which support long-term immune tolerance and survival. Alteration of this metabolic balance skews T cell differentiation toward pro-inflammatory effector phenotypes. For instance, the mTOR–AMPK pathway regulates T cell differentiation through bidirectional regulation of glycolysis and oxidative metabolism [53,54]. In chronic inflammatory diseases, continuous antigenic stimulation and cytokine exposure maintain immune cells in metabolically active phenotypes that metabolize high levels of glucose and glutamine. The continuous metabolic pressure contributes to cellular exhaustion and production of excessive inflammatory mediators. For example, synovial macrophages in rheumatoid arthritis and intestinal macrophages in inflammatory bowel disease display enhanced glycolytic flux and increased lactate secretion, which aggravate tissue inflammation and fibrosis. Thus, targeting metabolic checkpoints such as HIF-1α, mTOR, or glycolytic enzymes may offer strategies to modulate immune cell function therapeutically.

### 3.2. Metabolic Adaptations in Non-Immune Cells Within Inflammatory Microenvironment

Although immune cells have been the primary focus of immune-metabolism research, non-immune cells such as fibroblasts, endothelial cells, and epithelial cells also undergo metabolic changes during chronic inflammation. These cells contribute to tissue remodeling, angiogenesis, and fibrotic progression, and their metabolic characteristics strongly influence the inflammatory microenvironment. Inflammatory fibroblasts in diseases such as rheumatoid arthritis, idiopathic pulmonary fibrosis, and psoriasis exhibit a glycolytic phenotype similar to that of M1 macrophages. In the inflammatory synovium, fibroblast-like synoviocytes upregulate glucose transporter 1 (GLUT1) and hexokinase 2 (HK2), leading to increased glycolytic flux and lactate accumulation. The extracellular lactate lowers the pH of the local environment, promotes angiogenesis through vascular endothelial growth factor (VEGF) induction, and enhances leukocyte recruitment. These fibroblasts also display high levels of mitochondrial ROS and reduced OXPHOS efficiency, mediating metabolic demand to inflammatory mediator production [55,56]. This phenotype is largely driven by activation of the PI3K/Akt/mTOR signaling cascade, which enhances glycolytic enzyme expression and promotes inflammatory mediator production [57]. Epithelial cells exposed to chronic inflammatory stimuli such as cytokines or microbial products also exhibit metabolic remodeling. In intestinal epithelial cells, exposure to TNF-α or bacterial LPS reprograms energy metabolism from OXPHOS to glycolysis, leading to increased ROS and disruption of tight junction integrity. This metabolic reprogram compromises the epithelial barrier and exacerbates mucosal inflammation. Similarly, in airway epithelial cells, oxidative stress and mitochondrial dysfunction enhance production of inflammatory cytokines including IL-6 and granulocyte macrophage-colony stimulating factor (GM-CSF), which promote leukocyte infiltration and tissue injury [58,59,60]. Endothelial cells play a crucial role in regulating leukocyte migration and vascular homeostasis [61]. During chronic inflammation, endothelial cells adopt a glycolytic phenotype that supports proliferation and migration but reduces barrier integrity. Increased glycolysis in endothelial cells is mediated by activation of the 6-phosphofructo-2-kinase/fructose-2,6-bisphosphatase 3 (PFKFB3) enzyme, which enhances angiogenic extension and expression of adhesion molecules such as intercellular adhesion molecule-1 (ICAM-1). Inhibition of PFKFB3 or improvement of mitochondrial respiration in endothelial cells has been shown to reduce vascular inflammation in experimental models of atherosclerosis [62,63,64].

### 3.3. Metabolic Crosstalk in the Inflammatory Microenvironment

Interactions between immune and non-immune cells is also mediated by metabolic intermediates that function as signaling molecules. Lactate, succinate, fumarate, and itaconate are examples of metabolites that regulate inflammation. Lactate, produced by glycolytic cells, can act as a signaling molecule that modifies histones through lactylation and promotes transcription of genes involved in wound healing. Succinate accumulation stabilizes HIF-1α, promoting IL-1β expression, while fumarate can induce epigenetic changes that alter macrophage responses. Itaconate, derived from the TCA cycle intermediate cis-aconitate, exerts anti-inflammatory effects by activating NRF2 and inhibiting succinate dehydrogenase, thereby limiting ROS generation and cytokine production [65]. These findings demonstrate that metabolic products are not passive intermediates but active participants in regulating inflammatory responses. Dysregulation of metabolite signaling enhances inflammation by reprogramming gene expression and immune function. Targeting these metabolites or their associated enzymes may thus offer a new approach of therapeutic intervention in chronic inflammatory diseases. In particular, the itaconate, NRF2 and succinate, HIF-1α regulatory pathways represent contrasting metabolic pathways that determine inflammatory responses. Representative metabolic checkpoints and signaling pathways involved in immune and non-immune cell reprogramming are summarized in Table 2.

### 3.4. Therapeutic Implications of Metabolic Reprogramming

Recognition of metabolic control as a determinant of immune and tissue responses has prompted the exploration of metabolic pathways as drug targets. Agents that modulate glycolysis, fatty acid oxidation, or mitochondrial metabolism are being evaluated for their ability to re-establish immune balance. Pharmacological activators of AMPK, such as metformin, suppress pro-inflammatory cytokine production by enhancing mitochondrial respiration and limiting glycolysis. In contrast, inhibitors of mTOR, such as rapamycin, reduce effector T cell activity and promote regulatory T cell differentiation. Other approaches aim to restore NAD^+^ levels using precursors like nicotinamide riboside to support mitochondrial function and reduce oxidative stress [2,66]. Nutritional interventions, including caloric restriction, ketogenic diets, and exercise, can also modify systemic metabolism and influence inflammation. These strategies improve mitochondrial efficiency, enhance antioxidant defense, and modulate cytokine production. Although promising, metabolic interventions must consider the diversity of metabolic conditions among patients, as systemic alterations can affect multiple organs and cell types differently. Personalized approaches that account for individual metabolic signatures may optimize therapeutic efficacy.

## 4. Therapeutic Opportunities Targeting Mitochondrial and Metabolic Pathways

Mitochondrial dysfunction and metabolic reprogramming form critical mechanistic pathways in the pathogenesis of chronic inflammatory diseases. This realization has triggered development of therapeutic strategies aimed at re-establishing mitochondrial homeostasis, normalizing cellular metabolism, and thereby moderating chronic inflammatory responses. In this section, we explore four broad classes of therapeutic opportunities: (i) mitochondria-targeted antioxidants and redox modulators; (ii) modulators of metabolic checkpoints; (iii) nutritional, lifestyle and metabolic interventions; and (iv) developing advanced technologies such as mitochondrial transplantation, gene therapy and targeted delivery systems.

### 4.1. Mitochondria-Targeted Antioxidants and Redox Modulators

Given that excess mitochondrial ROS and consequent redox imbalance drive inflammatory signaling, antioxidants that localize to mitochondria represent a rational therapeutic approach. Traditional systemic antioxidants often are unable to reach the mitochondrial matrix in sufficient concentrations, or they lack specificity to the mitochondrial oxidative environment. Mitochondria-targeted antioxidants (MTAs) are designed with lipophilic cations or mitochondrial targeting sequences that promote accumulation driven by the negative mitochondrial membrane potential, thereby delivering the antioxidant component directly into the mitochondrial matrix. For example, compounds such as MitoQ, MitoVitE and SS-31 (Elamipretide) have demonstrated efficacy in reducing mitochondrial ROS production, improving mitochondrial membrane potential (ΔΨm), and attenuating downstream inflammatory cytokine release in preclinical models [6,67]. In chronic lung disease models such as chronic obstructive pulmonary disease (COPD), in vitro and ex vivo studies revealed that MTAs reduced macrophage mitochondrial ROS, improved mitophagy, and attenuated IL-1β secretion. The review by Fairley et al. documents that MTAs offer greater protective value compared to non-targeted antioxidants in respiratory inflammatory models [67]. In addition to lung disease, mitochondrial therapy has been proposed for metabolic inflammation, cardiovascular disease and autoimmune conditions. Zong et al. highlights that mitochondrial transplantation, mtDNA repair and mitochondrial-targeted small molecules are entering translational development efforts [6]. Nevertheless, there are a few of restrictions. Patient heterogeneity, safety, long-term effects, and the effectiveness of mitochondrial targeting are still challenges. In particular, the dual role of ROS in physiological signaling requires that antioxidant therapy does not suppress beneficial immune or metabolic processes. Accordingly, the future of MTAs may depend on better stratification of patients, biomarkers for mitochondrial dysfunction, and combination treatments with immunomodulators.

### 4.2. Modulators of Metabolic Checkpoints

Metabolic checkpoint regulators modulate the balance between glycolysis, oxidative phosphorylation, fatty acid oxidation and biosynthetic metabolism. These molecules act as mediators between cell metabolism and immune/tissue responses. Agents that modulate key signaling pathways such as AMP-activated protein kinase (AMPK), mechanistic target of rapamycin (mTOR), sirtuins (SIRT1/3), peroxisome proliferator-activated receptor-γ coactivator 1-α/β (PGC-1α/β) or NAD^+^-dependent deacetylases have developed as candidate drugs to reprogram metabolism and suppress inflammation. For example, activation of AMPK promotes mitochondrial biogenesis, enhances oxidative metabolism, and attenuates pro-inflammatory macrophage activation, whereas mTOR inhibition modulates immune cells from glycolytic effector phenotypes to regulatory phenotypes [68]. In metabolic inflammatory disease models, pharmacologic activation of AMPK (e.g., metformin) and supplementation with NAD^+^ precursors (e.g., nicotinamide riboside) have shown potential in reducing inflammatory mediators and improving mitochondrial functions. The review by Sinha et al. highlights that mitochondrial health can thus be modulated by metabolic checkpoint drugs [69]. The complexity of metabolic pathways, tissue-specific reactions, off-target effects, and long-term metabolic adaptability are examples of translational limitations. To advance these therapies, combination approaches integrating metabolic checkpoint modulation with immunotherapy or standard treatments may be required.

### 4.3. Nutritional, Lifestyle and Precision Metabolic Interventions

Because metabolism and mitochondrial function respond to exogenous factors, lifestyle and nutritional interventions represent non-pharmacologic or adjunctive strategies. Caloric restriction, intermittent fasting, ketogenic diet, exercise training and mitochondrial biogenesis promoters (e.g., PGC-1α activators) can improve mitochondrial respiratory capacity, reduce oxidative damage, remodel metabolic characteristics and attenuate inflammation. Li et al., review on metabolic inflammation outlines how metabolic disorders and inflammation converge on intersecting pathways and how lifestyle interventions can modulate these underlying mechanisms [70]. For example, regular aerobic exercise has been shown to enhance mitochondrial turnover (mitophagy and biogenesis), increase mitochondrial reserve capacity, and reduce markers of inflammation such as IL-6 and TNF-α. Nutraceuticals and supplements supporting NAD⁺ levels or mitochondrial antioxidant capacity also show potential, though comprehensive clinical trials are still limited. Precision metabolic interventions require measurement of individualized metabolic/mitochondrial biomarkers (e.g., mitochondrial DNA copy number, oxygen consumption rate of cells, metabolomic profile) to optimize the intervention strategy.

### 4.4. Developing Technologies: Mitochondrial Transplantation, Gene Therapy and Targeted Delivery Systems

With recent progress in the field, advanced technologies address mitochondrial dysfunction. Mitochondrial transplantation involves the transfer of viable mitochondria into injured or dysfunctional cells, aiming to restore mitochondrial integrity, respiratory capacity and reduce inflammatory signaling. Although still developing in inflammatory disease fields, preclinical studies in metabolic and ischemic injury models show improved mitochondrial function and reduced tissue damage [6]. Gene therapy approaches targeting mitochondrial generation regulators (e.g., PGC-1α/β, NRF1/2), mitochondrial maintenance control proteins (PINK1, Parkin), or mtDNA repair enzymes are currently being explored. Delivery of small interfering RNA or CRISPR/Cas9 constructs to modulate mitochondrial-specific genes has shown promise in model systems, though human translation remains challenging [71]. Nanocarrier and mitochondrial-targeted nanoparticle systems are being designed to deliver drugs specifically to the mitochondrial compartment (e.g., lipophilic cation linked small molecules, mitochondrial penetrating peptides). These delivery approaches may permit lower systemic doses while achieving high intra-mitochondrial drug concentrations and reducing off-target toxicity. Nevertheless, safety, targeting specificity, immunogenicity and long-term implications require further investigation [72,73]. Therefore, these developing technologies may transform the therapeutic approach from ameliorating symptoms to repairing the core metabolic and mitochondrial defects underlying chronic inflammation. Continued improvement of these approaches, in combination with patient-specific metabolic profiling, could accelerate their translation from experimental systems to clinical application. Current and developing therapeutic strategies targeting mitochondrial and metabolic pathways are summarized in Table 3.

## 5. Conclusions

Chronic inflammatory diseases result from a complex interaction between immune dysregulation, mitochondrial dysfunction, and metabolic reprogramming. Mitochondria act not only as bioenergetic centers but also as signaling regulators that regulate redox balance, apoptosis, and innate immune activation. Their dysfunction, through excessive ROS generation, mtDNA release, and defective mitophagy, amplifies inflammatory signaling and promotes tissue damage. Recent advances have revealed therapeutic opportunities targeting these mitochondrial and metabolic pathways. Mitochondria-targeted antioxidants, metabolic checkpoint modulators such as AMPK, mTOR, and sirtuins, and nutritional interventions show potential in restoring cellular homeostasis and attenuating inflammation. Developing strategies, including mitochondrial transplantation, gene-based modulation of mitochondrial homeostasis, and nanoparticle-mediated drug delivery, represent transformative approaches to repair the metabolic mechanisms driving inflammation. Integrating multi-omics profiling and metabolic phenotyping could also enable targeted therapeutic approaches based on individual mitochondrial and metabolic profiles. In summary, elucidating the molecular crosstalk between mitochondrial dysfunction and metabolic reprogramming provides a unifying perspective on chronic inflammation. Improving mitochondrial function and metabolic balance may enable long-term disease control in addition to treatment of clinical features. Integrated efforts in immunometabolism, systems biology, and clinical research will advance personalized therapies for inflammatory diseases.

## Figures and Tables

**Figure 1 cimb-47-01042-f001:**
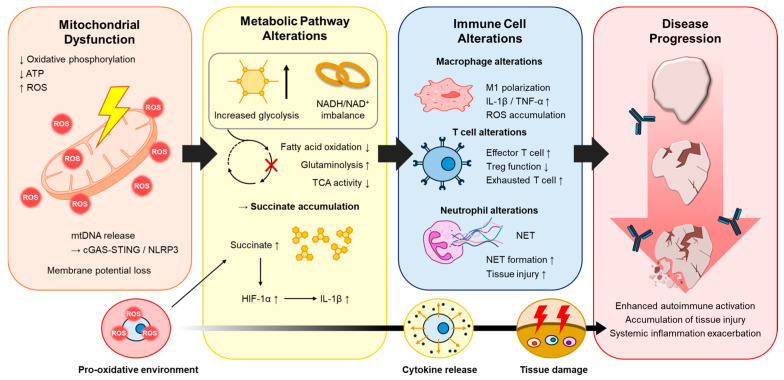
Mitochondrial dysfunction–induced metabolic alteration and immune dysregulation contributing to disease progression. Mitochondrial dysfunction reduces oxidative phosphorylation and ATP production while increasing ROS and releasing mtDNA, thereby activating the cGAS–STING and NLRP3 pathways. These events alter glycolysis, the TCA cycle, fatty acid oxidation, glutaminolysis, and NAD^+^ availability, generating metabolic programs that influence macrophages, T cells, and neutrophils. Macrophages exhibit M1 polarization with increased IL-1β, TNF-α, and ROS. T cells show expanded effector and exhausted populations with reduced Treg activity. Neutrophils display enhanced NET formation accompanied by tissue injury.

**Table 1 cimb-47-01042-t001:** Molecular Mechanisms Linking Mitochondrial Dysfunction to Chronic Inflammation.

Mechanistic Process	Molecular Regulators	Downstream Effect	Representative Diseases	Reference
Mitochondrial ROS overproduction	Complex I/III of ETC, NRF2–KEAP1, SOD2, PRDX3, GPX4	Oxidative stress, activation of NF-κB and MAPK signaling	Rheumatoid arthritis, COPD, SLE	[9,10]
mtDNA release	TFAM, Bax/Bak, VDAC, cGAS–STING, TLR9	Activation of type I IFN and inflammasome signaling	Sepsis, autoimmune disorders	[28,30]
Defective mitophagy	PINK1–Parkin, BNIP3/NIX, DRP1, MFN2	Accumulation of damaged mitochondria, NLRP3 activation	Atherosclerosis, type 2 diabetes	[31,35]
Altered mitochondrial dynamics	DRP1, OPA1, MFN1/2	Fragmentation, loss of membrane potential, ROS amplification	Neuroinflammation, metabolic syndrome	[8,32]

**Table 2 cimb-47-01042-t002:** Metabolic Checkpoints and Their Roles in Immune and Non-Immune Cell Reprogramming.

Metabolic Pathway	Regulatory Signaling Pathway	Functional Outcome	Cellular Targets	Inflammatory Context	References
Glycolysis	HIF1α with succinate pathway; PI3K/Akt/mTOR signaling pathway	Rapid ATP generation, IL-1β production, pro-inflammatory phenotype	M1 macrophages, activated T cells, fibroblasts	Rheumatoid arthritis, psoriasis	[49,54,55]
Fatty acid oxidation (FAO)	AMPK–PGC1β–PPARγ regulatory pathway; SIRT1 signaling	Mitochondrial biogenesis, anti-inflammatory polarization	M2 macrophages, Tregs, endothelial cells	Atherosclerosis, fibrosis	[2,50]
Amino acid metabolism	mTOR–ATF4–SIRT3 signaling cascade	T cell activation, oxidative balance	Effector T cells, epithelial cells	IBD, chronic airway inflammation	[53,54]
TCA cycle intermediates	Itaconate with NRF2 pathway; Succinate with HIF1α pathway	Anti- vs. pro-inflammatory metabolic switching	Macrophages, dendritic cells	Systemic inflammation, metabolic disease	[65,66]
Lactate metabolism	LDHA, HIF1α, and GPR81 signaling network	Histone lactylation, angiogenesis	Fibroblasts, endothelial cells	Tumor-associated inflammation, synovitis	[56,58,59]

**Table 3 cimb-47-01042-t003:** Therapeutic Strategies Targeting Mitochondrial and Metabolic Pathways in Chronic Inflammation.

Therapeutic Category	Primary Molecular Target	Mechanistic Action	Experimental/Clinical Evidence	Translational Challenges	References
Mitochondria-targeted antioxidants	MitoQ, SS-31, CoQ10 analogs; NRF2 and SOD2	Reduce mtROS, stabilize cardiolipin, restore ΔΨm	Improved mitochondrial function and reduced cytokine release in COPD and cardiovascular models	Limited targeting specificity; maintaining physiological ROS balance	[6,67]
AMPK activators/mTOR inhibitors	AMPK, mTORC1, SIRT3	Restore mitochondrial biogenesis, suppress glycolysis, promote oxidative metabolism	Metformin and rapamycin reduced inflammatory markers in preclinical and clinical studies	Systemic metabolic effects; dose-dependent adaptation	[68,69]
NAD^+^ boosters and Sirtuin activators	SIRT1/3, PGC1α	Enhance oxidative metabolism, improve redox balance	Nicotinamide riboside improved mitochondrial parameters in metabolic inflammation models	Long-term efficacy and tissue selectivity	[2,66]
Lifestyle/nutritional interventions	Caloric restriction, ketogenic diet, exercise	Enhance mitophagy and mitochondrial turnover; reduce systemic IL-6 and TNF-α	Clinical and animal studies show reduced inflammatory cytokines and improved mitochondrial function	Variability in adherence and metabolic heterogeneity	[66,70]
Gene or mitochondrial therapies	PGC1α, NRF2, PINK1, Parkin	Restore mitochondrial quality control; reprogram cellular metabolism at its source	Mitochondrial transplantation and gene modulation improved outcomes in ischemic and metabolic disease models	Delivery specificity, immune response, long-term stability	[1,71,72]

## Data Availability

No new data were created or analyzed in this study. Data sharing is not applicable to this article.

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
