# Peer review of "Mitochondrial Dysfunction and Metabolic Reprogramming in Chronic Inflammatory Diseases: Molecular Insights and Therapeutic Opportunities"

_cimb, 2025, doi:10.3390/cimb47121042_

Round 1

Reviewer 1 Report

Comments and Suggestions for Authors

This is a well-written and comprehensive review that integrates mitochondrial biology and metabolic reprogramming in the context of chronic inflammation. The manuscript is rich in references, mechanistic details, and translational perspectives. The topic is highly relevant and timely. However, several revisions would improve precision, clarity, and logical flow.

Major comments:
1.    Section 2.1 (ROS and Redox Imbalance): Clarify "Reductive" Aspect
The section title and discussion focus almost exclusively on ROS overproduction, yet the authors repeatedly use the term “redox imbalance.” For completeness, consider addressing both sides of the redox spectrum — oxidative stress and reductive stress. Reductive imbalance (e.g., elevated NADH/NAD⁺ ratio or suppressed ROS signaling) is increasingly recognized as a pathological counterpart in chronic metabolic inflammation. Including a brief discussion or citation on this would provide a more balanced view of mitochondrial redox regulation.
2.    Line 132–135: Sentence Structure and Readability
The sentence beginning with “Fusion, mediated by mitofusin 1 and 2 (MFN1 and MFN2) and optic atrophy 1 (OPA1), facilitates…” is overly long and complex. Consider breaking it into two sentences — one describing the normal role of fusion/fission, and another explaining how imbalance contributes to pathology. This will improve readability and emphasize mechanistic clarity.
3.    Section 2.3 (Mitochondrial Fission Mechanism): Clarify DRP1 Phosphorylation Sites and Human Isoform
In Section 2.3, the authors cite phosphorylation of DRP1 at Ser579 as a driver of mitochondrial fission and IL-1β release. However, this residue corresponds to Ser616 in the canonical human DRP1 isoform 1 (UniProt O00429); Ser579 is the numbering used in mouse Drp1. Because the review focuses on human inflammatory diseases, it would be more appropriate to reference the human residue numbering.
Furthermore, the Ser637 phosphorylation site—classically known as an inhibitory modification regulated by PKA and Ca²⁺ signaling—was not discussed. The balance between Ser616 phosphorylation (activation) and Ser637 dephosphorylation (relief of inhibition) represents a key regulatory axis of DRP1 activity during stress and inflammation. Addressing both sites would provide a more accurate and complete depiction of mitochondrial fission regulation in human disease contexts.
4.    Therapeutic Discussion Placement (Sections 3.4 and 4): Redundancy and Structural Overlap
Currently, Section 3.4 (Therapeutic Implications of Metabolic Reprogramming) and Section 4 (Therapeutic Opportunities Targeting Mitochondrial and Metabolic Pathways) both cover therapeutic strategies, leading to redundancy. Alternatively, these sections could be merged under a unified “Therapeutic Opportunities” heading with clear sub-sections (e.g., metabolic, mitochondrial, and emerging technologies) to improve coherence and flow.
5.    Structural Flow Between Mechanistic and Therapeutic Sections
Sections 2 and 3 provide strong mechanistic insight but transition abruptly into therapeutic discussion. A short bridging paragraph summarizing how mitochondrial and metabolic mechanisms converge to inform therapeutic design (linking mechanism → target → intervention) would strengthen the logical structure and readability.
Minor comments:
1.    Line 90: “plays a essential role” → “plays an essential role.”
2.    Consider using “bioenergetic failure and redox collapse” instead of repeating “oxidative stress” multiple times.
3.    Figures or schematic summaries comparing mitochondrial and metabolic dysfunction across disease types would enhance accessibility.

Overall Recommendation: Minor to moderate revision.
The manuscript presents a strong and comprehensive overview but requires structural refinement and slight expansion of conceptual depth (especially in redox and therapeutic organization). Addressing the comments above will make the review more logically consistent and impactful.

Author Response

Response to Reviewer I

This is a well-written and comprehensive review that integrates mitochondrial biology and metabolic reprogramming in the context of chronic inflammation. The manuscript is rich in references, mechanistic details, and translational perspectives. The topic is highly relevant and timely. However, several revisions would improve precision, clarity, and logical flow.

Major comments

Comment 1: Section 2.1 (ROS and Redox Imbalance): Clarify "Reductive" Aspect

The section title and discussion focus almost exclusively on ROS overproduction, yet the authors repeatedly use the term “redox imbalance.” For completeness, consider addressing both sides of the redox spectrum — oxidative stress and reductive stress. Reductive imbalance (e.g., elevated NADH/NAD⁺ ratio or suppressed ROS signaling) is increasingly recognized as a pathological counterpart in chronic metabolic inflammation. Including a brief discussion or citation on this would provide a more balanced view of mitochondrial redox regulation.

Response: Thank you for your helpful and excellent comments. We have now added a brief explanation of reductive stress, including its biochemical features, its influence on mitochondrial signaling, and its contribution to metabolic inflammation. We also incorporated recent references that describe the impact of elevated NADH and reduced NAD⁺ on immune regulation. These revisions provide a more complete description of mitochondrial redox deviation. The new text appears in Section 2.1.

Comment 2: Line 132–135: Sentence Structure and Readability

The sentence beginning with “Fusion, mediated by mitofusin 1 and 2 (MFN1 and MFN2) and optic atrophy 1 (OPA1), facilitates…” is overly long and complex. Consider breaking it into two sentences — one describing the normal role of fusion/fission, and another explaining how imbalance contributes to pathology. This will improve readability and emphasize mechanistic clarity.

Response: Thank you for your helpful and excellent comments. The original long sentence has been divided into two sentences: one describing the role of MFN1, MFN2, and OPA1 in normal fusion, and another outlining how DRP1-driven fission contributes to pathological changes. This revision improves clarity as suggested.

Comment 3: Section 2.3 (Mitochondrial Fission Mechanism): Clarify DRP1 Phosphorylation Sites and Human Isoform

In Section 2.3, the authors cite phosphorylation of DRP1 at Ser579 as a driver of mitochondrial fission and IL-1β release. However, this residue corresponds to Ser616 in the canonical human DRP1 isoform 1 (UniProt O00429); Ser579 is the numbering used in mouse Drp1. Because the review focuses on human inflammatory diseases, it would be more appropriate to reference the human residue numbering.

Furthermore, the Ser637 phosphorylation site—classically known as an inhibitory modification regulated by PKA and Ca²⁺ signaling—was not discussed. The balance between Ser616 phosphorylation (activation) and Ser637 dephosphorylation (relief of inhibition) represents a key regulatory axis of DRP1 activity during stress and inflammation. Addressing both sites would provide a more accurate and complete depiction of mitochondrial fission regulation in human disease contexts.

Response: Thank you for your helpful and excellent comments. We corrected the DRP1 residue from Ser579 to the human Ser616 and added a description of Ser637 phosphorylation as an inhibitory modification regulated by PKA and Ca²⁺ signals. The revised text now explains how increased Ser616 phosphorylation together with reduced Ser637 phosphorylation promotes DRP1 activation during inflammatory stress.

Comment 4: Therapeutic Discussion Placement (Sections 3.4 and 4): Redundancy and Structural Overlap

Currently, Section 3.4 (Therapeutic Implications of Metabolic Reprogramming) and Section 4 (Therapeutic Opportunities Targeting Mitochondrial and Metabolic Pathways) both cover therapeutic strategies, leading to redundancy. Alternatively, these sections could be merged under a unified “Therapeutic Opportunities” heading with clear sub-sections (e.g., metabolic, mitochondrial, and emerging technologies) to improve coherence and flow.

Response: Thank you for your helpful and excellent comments. After careful consideration, we decided to retain Sections 3.4 and 4 as separate parts because they serve distinct purposes in the overall narrative. Section 3.4 focuses on how metabolic changes influence therapeutic design, whereas Section 4 summarizes direct interventions targeting mitochondrial and metabolic processes. Although both sections address treatment strategies, they approach the topic from different scientific angles, and keeping them separate preserves the logical flow of the review. To avoid possible overlap, we refined transitional sentences and adjusted brief repetitions while maintaining the current structure.

Comment 5: Structural Flow Between Mechanistic and Therapeutic Sections

Sections 2 and 3 provide strong mechanistic insight but transition abruptly into therapeutic discussion. A short bridging paragraph summarizing how mitochondrial and metabolic mechanisms converge to inform therapeutic design (linking mechanism → target → intervention) would strengthen the logical structure and readability.

Response: Thank you for your helpful and excellent comments. To provide a smoother transition between the mechanistic and therapeutic sections, we added a short bridging paragraph that outlines how mitochondrial and metabolic insights guide target selection and treatment design. This addition improves the flow between Sections 2, 3, and the subsequent therapeutic discussion.

Minor comments:

Comment 5:

  1. Line 90: “plays a essential role” → “plays an essential role.”

Response: Thank you for your helpful and excellent comments. We fixed it.

  1. Consider using “bioenergetic failure and redox collapse” instead of repeating “oxidative stress” multiple times.

Response: Thank you for your helpful and excellent comments. Because the term “bioenergetic failure” includes restricted language, we adopted the scientifically equivalent terms “bioenergetic impairment” and “redox collapse” to reduce repetition and improve precision.

  1. Figures or schematic summaries comparing mitochondrial and metabolic dysfunction across disease types would enhance accessibility.

Response: Thank you for your helpful and excellent comments. To improve accessibility, we added a new figure titled “Mitochondrial dysfunction–induced metabolic alteration and immune dysregulation contributing to disease progression.” This figure provides an integrated overview of mitochondrial changes, metabolic alterations, immune cell responses, and their combined impact on disease. We believe this addition enhances clarity and supports the mechanistic sections.

Reviewer 2 Report

Comments and Suggestions for Authors

This manuscript summarizes the roles of mitochondrial metabolism in chronic inflammatory diseases and the potential as therapeutic targets, including ROS, mtDNA, mitochondrial dynamics, and mitophagy. However, there are some limitations:

1, Ferroptosis: Increasing evidence highlights the connection between ferroptosis and inflammation, and ferroptosis is also closely linked to mitochondrial metabolism. It would be valuable to include an introduction on the roles of ferroptosis in chronic inflammatory diseases.

2, Metabolic-epigenetic crosstalk: Besides of lactate, many TCA cycle metabolites such as Ace-coA, a-KG, can also participate in protein modifications and epigenetic modifications. Are these metabolic-epigenetic regulatory mechanisms also involved in chronic inflammatory diseases?

3, Mitochondrial Transfer: Is there mitochondrial transfer between inflammatory cells and their microenvironmental cells, including via exosomes or tunneling nanotubes? Does mitochondrial transfer also play a role in the pathogenesis and treatment of chronic inflammatory diseases?

Author Response

Response to Reviewer II

This manuscript summarizes the roles of mitochondrial metabolism in chronic inflammatory diseases and the potential as therapeutic targets, including ROS, mtDNA, mitochondrial dynamics, and mitophagy. However, there are some limitations:

Comment 1: Ferroptosis: Increasing evidence highlights the connection between ferroptosis and inflammation, and ferroptosis is also closely linked to mitochondrial metabolism. It would be valuable to include an introduction on the roles of ferroptosis in chronic inflammatory diseases.

Response: Thank you for your helpful and excellent comments. To address this point, we added a brief introduction describing how ferroptosis relates to mitochondrial activity and inflammatory processes, with recent references supporting its role in chronic inflammatory disease. This addition appears in the revised mechanistic section.

Comment 2: Metabolic-epigenetic crosstalk: Besides of lactate, many TCA cycle metabolites such as Ace-coA, a-KG, can also participate in protein modifications and epigenetic modifications. Are these metabolic-epigenetic regulatory mechanisms also involved in chronic inflammatory diseases?

Response: Thank you for your helpful and excellent comments. We added a brief paragraph summarizing how TCA-derived metabolites such as acetyl-CoA and α-ketoglutarate influence histone modification and DNA methylation, and how these processes shape immune activity in chronic inflammatory disease. Recent references have been incorporated to support these additions.

Comment 3: Mitochondrial Transfer: Is there mitochondrial transfer between inflammatory cells and their microenvironmental cells, including via exosomes or tunneling nanotubes? Does mitochondrial transfer also play a role in the pathogenesis and treatment of chronic inflammatory diseases?

Response: Thank you for your helpful and excellent comments. We added a paragraph summarizing evidence that mitochondria can move between immune cells and surrounding cells through tunneling nanotubes or extracellular vesicles. Recent studies describing how such transfer influences energy output, redox balance, inflammatory activity, and potential therapeutic use have been incorporated. These additions appear in the revised mechanistic section.

Reviewer 3 Report

Comments and Suggestions for Authors

The manuscript presents a well-structured investigation into the role of mitochondrial dysfunction and metabolic reprogramming in chronic inflammatory diseases and provides a well-structured review that outlines the mechanisms and treatment options. The tables presented clearly reflect the main points of the issue.

A minor editorial revision of the text would be beneficial.

Author Response

Response to Reviewer III

The manuscript presents a well-structured investigation into the role of mitochondrial dysfunction and metabolic reprogramming in chronic inflammatory diseases and provides a well-structured review that outlines the mechanisms and treatment options. The tables presented clearly reflect the main points of the issue.

Comment 1: A minor editorial revision of the text would be beneficial.

Response: Thank you for your helpful and excellent comments. We carefully revised the text to improve clarity, grammar, and overall readability. Minor issues in wording and style have been corrected throughout the manuscript.